# *Halanaerobium polyolivorans* sp. nov.—A Novel Halophilic Alkalitolerant Bacterium Capable of Polyol Degradation: Physiological Properties and Genomic Insights

**DOI:** 10.3390/microorganisms11092325

**Published:** 2023-09-15

**Authors:** Yulia Boltyanskaya, Tatjana Zhilina, Denis Grouzdev, Ekaterina Detkova, Nikolay Pimenov, Vadim Kevbrin

**Affiliations:** 1Winogradsky Institute of Microbiology, Research Center of Biotechnology of the Russian Academy of Sciences, 33, bld. 2, Leninsky Ave., Moscow 119071, Russia; julia_bol@rambler.ru (Y.B.); zhilinat@mail.ru (T.Z.); detkovkate@rambler.ru (E.D.); npimenov@mail.ru (N.P.); 2SciBear OU, Tartu mnt 67/1-13b, 10115 Tallinn, Estonia; denisgrouzdev@gmail.com

**Keywords:** *Halanaerobium*, polyol degradation, glycerol, ethanolamine, alkaliphilic microbial community

## Abstract

A search for the microorganisms responsible for the anaerobic degradation of osmoprotectants in soda lakes resulted in the isolation of a novel halophilic and alkalitolerant strain, designated Z-7514^T^. The cells were Gram-stain-negative and non-endospore-forming rods. Optimal growth occurs at 1.6–2.1 M Na^+^, pH 8.0–8.5, and 31–35 °C. The strain utilized mainly sugars, low molecular polyols, and ethanolamine as well. The G+C content of the genomic DNA of strain Z-7514^T^ was 33.3 mol%. Phylogenetic and phylogenomic analyses revealed that strain Z-7514^T^ belongs to the genus *Halanaerobium*. On the basis of phenotypic properties and the dDDH and ANI values with close validly published species, it was proposed to evolve strain Z-7514^T^ within the genus *Halanaerobium* into novel species, for which the name *Halanaerobium polyolivorans* sp. nov. was proposed. The type strain was Z-7514^T^ (=KCTC 25405^T^ = VKM B-3577^T^). For species of the genus *Halanaerobium*, the utilization of ethylene glycol, propylene glycol, and ethanolamine were shown for the first time. The anaerobic degradation of glycols and ethanolamine by strain Z-7514^T^ may represent a novel metabiotic pathway within the alkaliphilic microbial community. Based on a detailed genomic analysis, the main pathways of catabolism of most of the used substrates have been identified.

## 1. Introduction

Soda lakes are closed basins with a progressive evaporative concentration of inorganic salts, where sodium carbonates are prevailing, thereby providing a constantly alkaline pH of lake water. Soda lakes harbor diverse communities of haloalkaliphilic microbes, mostly prokaryotes that are well adapted to survive and grow in these extreme environments [1]. In addition to tropical and subtropical areas, they are widespread in the arid zone of Central Asia and, particularly, in southern Siberia. Similar to freshwater microbial communities, alkaline ones have a complete set of functionally significant microbes that are linked together via metabiotic (i.e., substances circulating extracellularly) interrelations [2,3]. Despite the good knowledge of the main pathways in alkaline communities [1,3,4], a number of trophic interactions remain unexplored to date. This may include the anaerobic decomposition of polyols.

Polyols are organic compounds which contain two and more hydroxyl groups and include liquid low molecular compounds ethylene glycol, 1,2-propanediol (propylene glycol), 1,3-propanediol, glycerol, and the others so-called sugar alcohols, solid substances starting with four-carbon compounds. Low molecular polyols are bulk chemical products and widely used in food, chemical, and pharmaceutical industries and, hence, can be regarded as organic pollutants. In addition to industry, the release of polyols into the environment may be due to biological activity. This is a matter of interest for reconstruction of intracommunity metabiotic pathways. The only exception here is ethylene glycol, for which, so far, the natural producers are unknown.

The occurrence of polyols in nature is quite wide, which is explained by the variety of their functions in the cells of organisms. As pointed out in [5], they can serve several roles in living organisms: act as carbohydrate reserves, as translocatory compounds, as a storage of reducing power, take part in osmoregulation and in coenzyme regulation. Glycerol is a well-known eukaryotic osmoprotectant that many, but especially aquatic, organisms accumulate to protect against osmotic stress [6,7]. A classic example, unicellular green algae *Dunaliella*, varies in its intracellular concentration of glycerol according to direct proportion of the extracellular salt concentration and, at saturation, can reach up to 50% of the total dry weight of the cell [8]. After dying off, the accumulated glycerol is released and becomes available to heterotrophs harbor the community. Wide distribution of *Dunaliella* spp. is shown for microbial communities of Kulunda steppe region where Tanatar soda lakes are located [9]. The latter are the object of our long-term research [10,11].

Propylene glycol is formed as a by-product in the course of anaerobic fermentations of sugars, mainly mono- and deoxy- sugars, rhamnose, and fucose [12,13]. The latter are structural fragments of glycosides and polysaccharides of terrestrial plants, algae, and fungi. Solid polyols, erythritol, mannitol, sorbitol, and the others are widely distributed in fungi [14], especially in halotolerant and halophilic ones [15], where they function as osmoprotectants similar to glycerol and sometimes together with glycerol. Once in the anaerobic conditions of a salt (and/or alkaline) lake, they also become available for utilization by heterotrophic microflora.

In saline conditions, there are many taxonomically diverse groups of bacteria capable of anaerobic fermentation. One of them belongs to the genus *Halanaerobium* (*Halanaerobiaceae*, *Halanaerobiales*, *Clostridia*, *Bacillota*). The type species of the genus was isolated from the deep sediments of Great Salt Lake, Utah, USA, and was effectively published in 1983 as *Halanaerobium praevalens* GSL^T^ [16]. It was later validly published and by now, according to the web resource LPSN (https://lpsn.dsmz.de), the genus includes 10 validly published species and one effectively published species ‘*H. hydrogeniformans*’ isolated out of haloalkaline Soap Lake, WA, USA [17]. The detailed characteristics of the species of the genus was collected in Bergey’s Manual [18] by Oren and the original publications cited there. 

While searching for glycerol degrading bacteria from soda lakes, we focused on anaerobic conditions, since in this case, decomposition products can be used by other members of the microbial community, giving rise to the corresponding trophic chains. Here, we described a novel representative of genus *Halanaerobium*, *H. polyolivorans* Z-7514^T^, capable of anaerobic utilization of hydroxyl compounds, i.e., glycols, glycerol, sugar alcohols, and ethanolamine as well.

## 2. Materials and Methods

### 2.1. Enrichment and Isolation

The characteristics of the natural sample, the composition of the media, and the techniques for their preparation have been described previously [11] except that betaine was replaced with 3 g/L of glycerol. For routine transfers and maintenance, after finding pH and Na^+^ optima, the following optimal medium was compiled (g/L): glycerol, 3; NaHCO_3_, 8; Na_2_CO_3_, 1.5; NaCl, 100; NH_4_Cl, 0.25. Other constituents were the same as for the enrichment medium. Apart from mineral constituents, the medium was supplemented with 0.1–0.2 g/L of yeast extract as an essential additive. pH value of optimal medium was 8.55.

### 2.2. Morphological Characterization

Cell morphology was examined using an Axio Imager D1 light microscope (Zeiss, Oberkochen, Germany) equipped with a phase-contrast unit. Electron micrographs were obtained by using a JEOL model JEM-100C transmission electron microscope and negative staining with 2% (*w*/*v*) sodium phosphotungstate. Gram reaction was determined using the Gram-staining kit (Deltalab, Barcelona, Spain).

### 2.3. Physiological Characterization

Growth of the strain was monitored by optical density at 600 nm measured directly in Hungate tubes using a Unico 2100 (Dayton, NJ, USA) spectrophotometer or by hydrogen production. Physicochemical parameters of growth (pH, salinity, and temperature ranges; sodium, chloride, and carbonate requirement) were determined as described previously [11] with glycerol as a substrate. The optimal medium was used for testing of substrates and electron acceptors. For media with polyols and ethanolamine, vitamin B_12_ (125 µg/L) was added. Substrates were tested at concentration of 3 g/L except for polymers (2 g/L). Thermally unstable in alkaline conditions substances, such as sugars, were added into the medium after sterilization. Most electron acceptors (sulfate, thiosulfate, DMSO, nitrate, pyruvate, fumarate, TMAO, acetone, acetoin, crotonate, and AQDS) were tested at concentration of 10 mM each except for betaine (20 mM), nitrite (5 mM), sulfite (5 mM), and sulfur (10 g/L). Tolerance for ethylene and propylene glycols was tested in the range of 25 to 200 mM.

### 2.4. Analytical Assays

Sugars, polyols, and organic acids were assayed using a Stayer HPLC chromatograph (Aquilon, Moscow, Russia) equipped with refractometric and UV detectors connected in series. Separations were carried out on Aminex HPX-87H column (Bio-Rad, Hercules, CA, USA), operated isocratically with 5 mM H_2_SO_4_ as eluent at flow rate 0.6 mL/min. Hydrogen was quantified on a Crystal 5000.2 GC chromatograph (Chromatek, Yoshkar-Ola, Mari El, Russia) equipped with a glass column (1 m × 3 mm) filled with Carboxen 1000 (Supelco, Bellefonte, PA, USA) operating at 140 °C, and a thermal conductivity detector at 180 °C. The carrier gas was argon at 40 mL/min. Ethanolamine was assayed by colorimetrically by the OPA method specific for primary amino groups [19]. The formation of sulfide was tested by the methylene blue formation reaction [20]. Ammonium was determined with Nessler’s reagent after isothermal microdistillation of free ammonia. 

### 2.5. Genome Sequencing and Bioinformatic Analyses

The 16S rRNA gene sequence for strain Z-7514^T^, which is nearly complete at 1435 bp, was derived using primers 8-27f (5′-AGAGTTTGATCCTGGCTCAG-3′) and 1492r (5′-TACGGYTACCTTGTTACGACTT-3′) as described in [21]. The 3730 DNA Analyzer from Applied Biosystems, Waltham, MA, USA, was used for sequencing with the Big Dye Terminator reagent kit, version 3.1. The GenBank/EMBL/DDBJ recorded the sequence for strain Z-7514^T^ under the accession number OK643887. To taxonomically classify the strain, a phylogenetic tree was constructed using 16S rRNA gene sequences from strains of the genus *Halanaerobium* and adjacent genera. The sequences were first aligned using MUSCLE [22], and the maximum likelihood tree was constructed using the GTR+F+I+G4 model proposed by ModelFinder [23] and implemented in IQ-Tree [24]. Branch supports were determined using 10000 ultrafast bootstraps [25]. 

Isolation of genomic DNA, instrumentation for sequencing, and software for quality check and assembly was performed as reported earlier [11]. A total of 4,304,511 paired-end reads were obtained from strain Z-7514^T^. Phylogenomic analysis of strain Z-7514^T^ was performed using a concatenated alignment of 120 single-copy phylogenetic marker genes obtained using the software GTDB-Tk version 1.0.2 [26]. Maximum likelihood phylogenomic tree was inferred using IQ-Tree [24] with model recommended by ModelFinder [23] and branch support was estimated using UFBoot2 [25]. AAI values were determined using CompareM version 0.0.23, available at https://github.com/donovan-h-parks/CompareM (accessed on 9 September 2023). The default parameters for blastp were used, which include an e-value of ≤ 0.001, a minimum percent identity of ≥30%, and an alignment length of ≥70%. POCP values were calculated using the script runPOCP.sh [27], which follows the methodology previously described in [28]. The digital DNA–DNA hybridization (dDDH) and average nucleotide identity (ANI) values were performed by Genome-to-Genome Distance Calculator 3.0 (GGDC) [29] and FastANI 1.3 [30], respectively. The dDDH results were based on recommended formula 2 (identities/HSP length). 

Identification of protein-coding sequences and primary annotation were performed using the NCBI Prokaryotic Genome Annotation Pipeline (PGAP) [31]. To identify the Clusters of Ortholog Groups (COGs) [32] and Pfam domains [33] in the predicted protein sequences, the reCOGnizer 1.9.1 tool was used [34]. Gene search and overview of the whole-genome were performed using the RAST annotation server [35] and the BlastKOALA annotation tool at the KEGG database [36]. Gene maps were constructed using the GizmoGene (http://www.gizmogene.com) web resource. A search for appropriate enzymes was performed at the BRENDA database [37].

## 3. Results

### 3.1. Isolation

Strain Z-7514^T^ was isolated as a contaminant in the course of isolation of betaine-degrading *Halonatronomonas betaini* Z-7014^T^ [11]. Although the isolation medium for strain Z-7014^T^ did not contain glycerol, another 16S RNA was consistently present in betaine grown Z-7014^T^ as detected by molecular methods. We hypothesized that a contaminant used small amounts (50 mg/L) of yeast extract, which was necessary for the growth of strain Z-7014^T^. Success in the separation of strains was achieved by plating on a medium without betaine, but with an increased content (5 g/L) of yeast extract as a sole substrate for both strains. Visually, grown colonies were very similar and could only be distinguished by molecular methods. Picked up colonies were transferred to liquid medium with glycerol and those that had growth were plated again on solid medium with glycerol. To be sure in purity, several rounds of “liquid culture—solid culture” on glycerol containing medium were performed. The strain was designated Z-7514^T^ and selected for further characterization.

### 3.2. Colonies and Cell Morphology

On the fifth day of incubation in solid medium with glycerol, the strain formed colonies reached 0.5–1.5 mm in diameter. In the depth of agar, they were disc-shaped, milky-white opal colored with a granular structure and a denser protrusion in the center. The deep colonies were about 0.5–1.0 mm in diameter. The surface colonies were round and slightly larger (1.0–1.5 mm in diameter) than those in deep and had a smooth transparent edge, radiating with a bluish-pink color and white center. Cells from deep and surface colonies were morphologically uniform and genetically homogeneous.

In optimal liquid medium, cells were small, straight, or slightly curved rods 1–2 µm in length and 0.4–0.8 µm in width, sometimes close to cocci in shape, occurring singly or in short chains of 2–3 cells (Figure 1A) and less often up to 5–6 cells. 

Reproduction occurs by binary fission. Motility was observed very rarely, only in single cells and not at every viewing. Endospores were not observed and the cells were Gram-stain negative, which is typical for species of *Halanaerobium* [18]. In cells grown on glycerol-containing medium, the electron micrograph revealed the presence of bacterial microcompartments (metabolosomes, in this case), which is typical for polyol grown organisms (Figure 1B).

### 3.3. Phylogenetic and Genomic Characterization

Phylogenetic analysis based on the 16S rRNA gene sequences revealed that Z-7514^T^ clustered with members of the genus *Halanaerobium* (Figure 2).

The closest to strain Z-7514^T^ was turned ‘*H. hydrogeniformans*’ with 98.6% similarity. According to the tree, the 16S rRNA gene sequence of Z-7514^T^ shared 97.5, 97.5, and 97.2% similarity to the sequences with *H. kushnerii* ATCC 700103^T^, *H. saccharolyticum* subsp. *saccharolyticum* Z-7787^T^, and *H. praevalens* GSL^T^, respectively. A high similarity to the closely related sequences supports the identification of the strain Z-7514^T^ as belonging to a genus *Halanaerobium*. In addition to Z-7514^T^, the other 16S rRNA gene sequences were clustered with the same genus with a similarity of >98%. They were found in saline alkaline intertidal soils of the Gulf of Cambay, India (clones JX240593-JX240713) [38], and haloalkaline laboratory biogas reactor using algae *Spirulina* as a substrate (clone KR476501) [39].

To clarify the taxonomic position of strain Z-7514^T^ according to the modern standards, a complete genomic sequencing was performed. The final assembled 2,522,622-bp-long genome comprised 53 scaffolds, with an N_50_ value of 125,414 bp, a G+C content of 33.3%, and a coverage of 467.9×. The 16S rRNA gene sequence out of genome was identical to the sequence obtained by PCR. The inferred phylogenomic tree also indicates that the strain belongs to the genus *Halanaerobium* (Figure 3). The detailed genome statistics of strain is shown in Table 1.

The G+C content of strain Z-7514^T^ was within the range of 27–37 mol% characteristic for the species of *Halanaerobium* [18]. The genomic indices of strain Z-7514^T^ in relation to species of the genus *Halanaerobium* whose genomes were accessible are shown in Table 2. All of them were below the cut-off value 70% for dDDH [40,41] and of 95% for ANI species identity [41]. The AAI and POCP values also were below the statistical threshold for species delineation, but not for genera [28,42], and confirmed that strain Z-7514^T^ represents a new species in the genus *Halanaerobium*.

### 3.4. Physiological Properties

#### 3.4.1. Physicochemical Characteristics of Growth

Strain Z-7514^T^ appeared to be a moderate halophile with growth range at 0.6–3.9 M Na^+^ and optimum at 1.6–2.1 M Na^+^. The optimal pH for growth was 8.0–8.5 while the growth interval was 6.7–10.1. Due to the lower limit of the pH range less than 7, the strain should not be considered as a true alkaliphile but alkalitolerant instead. Strain Z-7514^T^ possessed an obligate requirement in sodium and chloride ions, and it could grow in three consecutive transfers on a carbonate-free medium. No growth was found in chloride-free medium buffered with bicarbonate/carbonate, equimolar to sodium. The requirement for chloride was characteristic of *Halonatronomonas betaini* [11] and some other bacteria isolated by us from Tanatar lakes. In relation to temperature, strain Z-7514^T^ was mesophilic, at optimal pH and salinity, it grew at 14–51 °C with optimum at 31–35 °C. The strain did not tolerate oxygen being unable to grow under cotton plug, but it was able to grow in anaerobic medium lacking reducing agent.

#### 3.4.2. Substrates, Electron Acceptors and Phenotypical Comparisons

Strain Z-7514^T^ has mostly a fermentative type of metabolism, particularly saccharolytic. No protein or proteinaceous substrates were used except for yeast extract. It grew on a range of carbohydrates, a complete list of which is given in the species description. The other used substrates included N-acetyl-D-glucosamine, acetoin, pyruvate, meso-erythritol, mannitol, glycerol, glycerol 3-phosphate, yeast extract, ethylene glycol, propylene glycol, and ethanolamine. Some substrates, unusual for *Halanaerobium* species (polyols, except for glycerol, and ethanolamine), were decided to be added to the list of tested substrates in the course of genomic analysis (see below), when the corresponding genes were found. Depending on the substrate, fermentation products included mainly acetate, hydrogen, and often but not always lactate. Propylene glycol fermented to propionate, n-propanol, and hydrogen; no acetate was found. Ethylene glycol fermented to acetate and hydrogen only. Of any of the substrates, neither formate nor ethanol was found. For some substrates, the fermentation products were quantified (Table 3).

The growth on ethanolamine turned out to be very slow but reproducible in five successive transfers. The maximal optical density about 0.03 was reached by the 30th day of growth. The addition of vitamin B_12_ increased optical density twice; maximum was already achieved by the 8th day of growth. This is consistent with the data that the use of ethanolamine both as a carbon or as a nitrogen source requires a derivative of vitamin B_12_ (Ado-B_12_, adenosylcobalamine) which is a cofactor of ethanolamine lyase [43,44].

No electron acceptors, except elemental sulfur, were used. Sulfur was reduced to sulfide (about 7 mM is formed). To date, the ability to reduce sulfur was reported only for two *Halanaerobium* species, *H. saccharolyticum* subsp. *saccharolyticum* [45] and *H. congolense* [46]. The main phenotypic features distinguishing the strain Z-7514^T^ from close species of *Halanaerobium* with validly published names are shown in Table 4.

Differences from closely related species included a different set of used substrates, the pH optimum was shifted to the alkaline side, and the absence of formate and ethanol among the fermentation products.

#### 3.4.3. Utilization of Polyols

Despite the strain being isolated on glycerol, the maximum utilization efficiency was noted for propylene glycol, which was almost completely consumed by the 6th day of incubation (Appendix A). On the contrary, the utilization of glycerol was rather slow and amounted to only 37% after 30 days of incubation. Ethylene glycol occupied an intermediate position. The most probable explanation of this phenomenon is that the propanediol dehydratase encoded in genome has different affinity towards different polyols as transporters of facilitated diffusion, neither glycerol nor propylene glycol were found (see below), and polyols were directly delivered to the dehydratase. The stimulating effect of vitamin B_12_ in the course of growth on propylene glycol was manifested by an increase in the biomass yield (increase in OD), but not in the growth rate. Interestingly, B_12_ had no effect on the ethylene glycol grown culture.

Considering the possibility of using the strain Z-7514^T^ to remove ethylene and propylene glycol from the environment, it was decided to test the strain towards elevated concentrations of polyols. Growth occurred on both polyols in the range of 25–200 mM given initial concentrations. Residual contents of polyols were measured for propylene glycol in stationary step of growth (10 days) and for ethylene glycol in 30 days of incubation. The utilization value calculated as a difference between initial and final concentration and was plotted versus initial concentration (Appendix A). As can be seen, the utilization of ethylene glycol gradually increased with an increase in its initial concentration in the medium, although depletion was uncomplete. On the contrary, the utilization of propylene glycol at first increased with an increase in its concentration, reached its maximum at 50 mM, and then further decreased. Unlike ethylene glycol, exhaustion was almost complete at 25 mM but, hereinafter, residual concentration increased. In both cases, strain Z-7514^T^ showed a good tendency to reduce glycol content at elevated concentrations.

### 3.5. Physiology and Functional Genes

The general functional annotation of the genome of strain Z-7514^T^ is presented in Figure 4. Complete results are presented in Appendix A.

Positive tests for substrates correlated with the genes found in course of genome analysis. Particularly, the genome contained the genes encoding all enzymes of the Embden–Meyerhof–Parnas (EMP) pathway except for a canonical hexokinase (EC 2.7.1.1), which was substituted for glucokinase (EC 2.7.1.2) encoded by *glk* operon and fructokinase (EC 2.7.1.4) encoded by *scr*K. Also, all four enzymes of the Leloir pathway for galactose utilization encoded by *gal*M, *gal*K, *gal*T, *gal*E were present. Pentose phosphate pathway (PPP) was represented only by the non-oxidative branch, since the operon *pgl* encoding 6-phosphogluconolactonase was absent. In addition to common sugars, the utilization of the other substrates also correlated with the presence of appropriate genes. For mannitol, an operon *mtl*ABC encoding a mannitol phosphotransferase system along with mannitol-1-phosphate dehydrogenase (EC 1.1.1.17, *mtl*D) was found. The product of the latter, D-fructose 6-phosphate, fed the EMP pathway. A similar phosphotransferase system was also available for N-acetyl-D-glucosamine (EC 2.7.1.193, *nag*E). The genome of Z-7514^T^ lacked both specialized erythritol catabolic genes *ery* firstly discovered in the *Brucella* species [48] and erythritol kinases EC 2.7.1.27 and EC 2.7.1.215. A genome and enzyme search showed that two enzymes may possibly be responsible for the catabolism of erythritol, namely glycerol dehydrogenase (EC 1.1.1.6, *gld*A) [49] and L-iditol (D-sorbitol) 2-dehydrogenase (EC 1.1.1.14, *gut*B). For the last one, the transformation of erythritol to erythrulose for enzyme isolated from *Gluconobacter suboxidans* [50] was observed. Its activity toward erythritol was 1.7 times higher than sorbitol. For Z-7514^T^, however, when sorbitol was added, there was neither growth nor the formation of any products. 

Three low molecular polyols were used by strain Z-7514^T^, namely ethylene glycol, propylene glycol, and glycerol. The anaerobic degradation all of them is controlled by a known *pdu* gene cluster [51,52,53] (not always in case of glycerol, see below), which is found in genome. For clarity, this cluster of Z-7514^T^ was aligned with the well-characterized *pdu* proteins of *Salmonella enterica* subsp. *Enterica* serovar *Thyphimurium* LT2 (Figure 5A). 

The *pdu* cluster of Z-7514^T^ encodes for seven proteins with putative enzymatic function in polyol degradation, seven putative structural proteins (metabolosomes), one protein with a putative regulatory function (*pdu*GH), and three proteins with unknown/unclear (*eut*J) function. The cluster, as well as in *S. enterica*, contained a core operon *pdu*CDE encoding three subunits of propanediol dehydratase (EC 4.2.1.28) along with the genes that ensure further reactions of polyol degradation. Also, the cluster included the genes encoding proteins of metabolosome (Figure 1B). 

In a general way, the *pdu* cluster of Z-7514^T^ was somewhat similar with that of *S. enterica* but had some differences. Homologs of the genes encoding a facilitated diffusion of polyols into the cell (*pdf* or *glp*F), propanol dehydrogenase (*pdu*Q), phosphate propanoyltransferase (EC 2.3.1.222, *pdu*L), and propionate kinase (EC 2.7.2.15, *pdu*W) were absent in the gene cluster of Z-7514^T^. The last three genes are involved in the propylene glycol degradation pathway in enteric bacteria [54]. Obviously, their functions are carried out by other enzymes.

The ability to use ethanolamine by strain Z-7514^T^ was due to the presence of ethanolamine ammonia lyase (EC 4.3.1.7) encoding by *eut*BC operon. The enzyme split ethanolamine into acetaldehyde and ammonia. As for polyols, *S. enterica* also served as a model object for comparison. The *eut* cluster of Z-7514^T^ was aligned with the homological one, and was present in *S. enterica* (Figure 5B). Unlike *S. enterica*, the *eut* genes of Z-7514^T^ were not concatenated in one operon but distributed among the *eut* and *pdu* clusters. The *eut* cluster of Z-7514^T^ turned out to be significantly less than in *S. enterica* and encoded for two subunits of ethanolamine ammonia lyase, two structural proteins (*eut*L, proteinpgf), and one protein with a putative regulatory function (*eut*A). The other *eut* genes were located in the *pdu* cluster and included mainly the BMC proteins except for *eut*P which encodes acetate kinase functionally similar but structurally differed from *ack*A [55]. It is noteworthy that the key ethanolamine catabolite gene encoding acetaldehyde dehydrogenase (*eut*E) was absent, but growth proceeded, therefore, similar to enzymes of the *pdu* cluster, and its function was performed by the other enzyme.

## 4. Discussion

Siberian soda lakes, including the Tanatar lakes, are inhabited by bacteria, the diversity of which is not inferior to those living in freshwater lakes. A detailed metagenomic study found all representatives required to form an autonomous microbial community, namely prime producers, hydrolytic microorganisms, secondary utilizers, and mineralizers like sulfate reducers and methanogens [56]. It is noteworthy that the *Halanaerobiales*-related OTUs, particularly the genus *Halanaerobium,* were the second most abundant among the filum *Bacillota* (*Firmicutes* in the reference), and we confirmed these results by isolating the representative of the genus.

The data obtained from genomic analysis made it possible to draw a hypothetical scheme for the metabolism of some substrates of strain Z-7514^T^ (Figure 6). 

Sugars are catabolized via the EMP and PPP pathways, yet ribose entry into the PPP pathway remains unclear, since ribokinase (EC 2.7.1.15), which is common to most bacteria growing on ribose, is absent. The initial steps of polyol degradation are carried out by nonspecific B_12_-dependent propanediol dehydratase encoded by operon *pdu*CDE. It performs the dehydration step with formation of the corresponding aldehyde. The product of dehydration depends on particular polyol: acetaldehyde, propanal, and 3-hydroxypropanal from ethylene glycol, propylene glycol, and glycerol, respectively. Then, three-carbon aldehydes are reduced to the corresponding alcohols by 1,3-propanediol dehydrogenase. For glycerol, in addition to the reductive way described above, the oxidative way also exists starting with glycerol dehydrogenase and glycerol kinase and further feeds the EMP pathway. Not all glycerol-utilizing bacteria have a reductive way and those bacteria do not possess propanediol or glycerol dehydratase and, respectively, do not produce 1,3-propanediol.

For propylene glycol, its entry into the EMP pathway through glycerol dehydrogenase is also potentially possible because, for example, the enzyme from *Cellulomonas* sp. had almost the same activity to propylene glycol as for glycerol [57], but it did not work in case of Z-7514^T^, since acetate was not formed during growth on propylene glycol. The lack of phosphate propanoyltransferase and propionate kinase, encoded by genes *pdu*L and *pdu*W, is probably replenished by phosphate acetyltransferase (EC 2.3.1.8, *pta*) and acetate kinase (EC 2.7.2.1, *ack*A), respectively, as was shown earlier [54]. For acetate kinase, activity data in relation to propionate vary in the literature, but can reach 93% for *Cereibacter* (former *Rhodobacter*) *sphaeroides* [58]. In addition, an operon *fak*AB encoding nonspecific fatty acid kinase (EC 2.7.2.18) was found in the genome and it can also contribute to the formation of propionate from propionyl-CoA. Fewer data are available to substrate specificity of phosphate acetyltransferase but, for example, for *Thermotoga maritima*, phosphate acetyltransferase accepted propionyl-CoA with 60% activity lower than acetyl-CoA [59]. The functions of the missing propanol dehydrogenase (*pdu*Q) are probably performed by alcohol dehydrogenase (EC 1.1.1.1, *adh*) or 1,3-propanediol dehydrogenase (EC 1.1.1.202, *dha*T). The last one is less probable, because for the enzyme from *Citrobacter freundii*, its activity towards propanal was five times worse compared to 3-HPA [60]. Of course, further biochemical work is required to confirm our assumptions.

The anaerobic catabolism of ethylene glycol and ethanolamine proceeds expectedly via acetaldehyde, since alternative pathways are currently unknown. The low yield of biomass on ethanolamine, in addition to the need for B_12_, may be due to the absence of one of the genes required for the assembly of the metabolosome. These intracellular protein formations encapsulate the enzymes dealing with the toxic intermediates, aldehydes, and widely distributed among polyol and ethanolamine degrading microorganisms [61]. The formation of the structure encoding at least five *eut* genes [62], of which the genome of the strain contains only four and *eut*K, is absent and it may not be enough to build a complete icosahedral structure. This gene encodes one of the major shell constituents of a functionally complex ethanolamine utilization microcompartment [63]. The natural precursor of ethanolamine is phosphatidylethanolamine, a lipid component of membranes characteristic of both bacterial and eukaryotic cells. Under the action of phosphodiesterases, this lipid is split into glycerol 3-phosphate and ethanolamine [64]. Both products can be utilized by Z-7514^T^. Prokaryotic and eukaryotic cells are always present in modern microbial communities; therefore, the ethanolamine pathway of biomass degradation, like the proteolytic one, can be classified as a permanent one. 

At the moment, it is difficult to say how widespread the use of polyols and ethanolamine is among species of *Halanaerobium*. To date, complete genomes deposited in the publicly available databases are available for only half of the 10 validly published species plus effectively published ‘*H. hydrogeniformans*’ genome, which is not enough to generalize. Nevertheless, we found the *pdu* genes only in two genomes, namely *H. saccharolyticum* subsp. *saccharolyticum* and ‘*H. hydrogeniformans*’. At the same time, all six sequenced species possessed *glp*K, glycerol kinase, and five species—*gld*A, glycerol dehydrogenase, although the ability to grow on glycerol, with the exception of the above-mentioned species, was shown only for *H. kushneri* [47]. This phenomenon requires further study. The *eut*BC operon was found in three genomes, namely *H. congolense, H. praevalens,* and ‘*H. hydrogeniformans*’ but the authors of the publications have not tested whether these species can grow on ethanolamine. So, the ability of species of the genus *Halanaerobium* to use polyols and ethanolamine may be underestimated.

In summary, based on phylogenetic and phenotypic data, we assume that strain Z-7514^T^ represents a novel species in the *Halanaerobium* genus with a proposed name *Halanaerobium polyolivorans* sp. nov.

## 5. Description of *Halanaerobium polyolivorans* sp. nov.

*Halanaerobium polyolivorans* (po.ly.o.li.vo’rans. N.L. neut. n. *polyol*, an organic compound containing multiple hydroxyl groups; L. inf. v. *vorare*, to eat; N.L. part. adj. *polyolivorans*, polyol-eating).

The cells are small straight or slightly curved rods that are 0.4–0.8 µm wide and 1–2 µm long, sometimes close to cocci. Growing cells appear singly, in pairs, or as short chains. They are Gram-stain-negative, spores are not observed, and are catalase- and oxidase-negative. Chemoheterotrophic growth occurs under anaerobic conditions. They contain moderate levels of halophile, alkalitolerant, and mesophile. Growth occurs at 0.6–3.9 M Na^+^ (optimum, 1.6–2.1 M Na^+^), at pH 6.7–10.1 (optimum, pH 8.0–8.5), at 14–51 °C (optimum, 31–35 °C); they obligately depend on Na^+^ and chloride ions, but do not need carbonates in medium. The following substrates support growth: D-fructose, D-ribose, D-glucose, D-mannose, D-xylose, D-galactose, cellobiose, trehalose, sucrose, maltose, N-acetyl-D-glucosamine, acetoin, pyruvate, meso-erythritol, mannitol, glycerol, glycerol 3-phosphate, yeast extract, ethanolamine, ethylene glycol, and propylene glycol. No growth on melibiose, melezitose, raffinose, lactulose, L-rhamnose, lactose, L-fucose, L-sorbose, L-arabinose, xylitol, inositol, sorbitol, dulcitol, succinate, gluconate, formate, acetate, malate, fumarate, lactate, glycolate, methanol, ethanol, 1-propanol, 2,3-butandiol, trypticase, tryptone, peptone, soytone, casamino acids, and betaine. Caseinate, dextran, starch, and xylan are not hydrolyzed. They reduce sulfur to sulfide. The G+C content from the DNA is 33.3 mol %. The type-strain, Z-7514^T^ (=KCTC25405^T^ = VKM B-3577^T^), was isolated from sediments of a collector at Tanatar III soda lake, Altai region, Russian Federation.

## Figures and Tables

**Figure 1 microorganisms-11-02325-f001:**
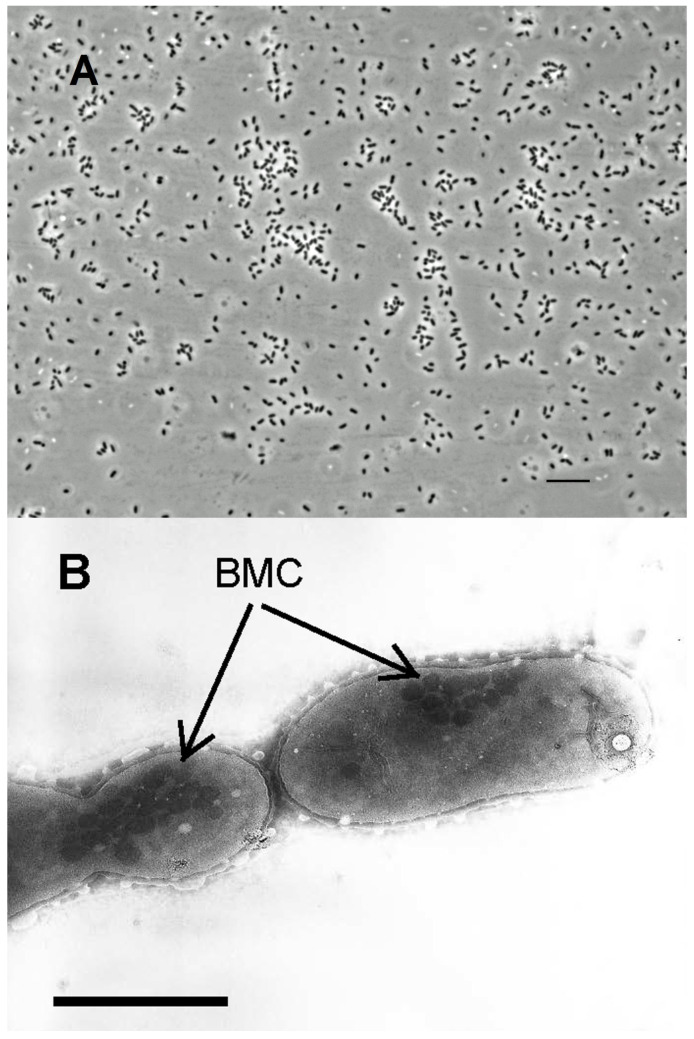
View of strain Z-7514^T^ grown on glycerol: (**A**) phase-contrast light microscope image. Scale bar, 10 µm; (**B**) electron micrograph image. Arrows point to bacterial microcompartments (BMC). Scale bar, 1 µm.

**Figure 2 microorganisms-11-02325-f002:**
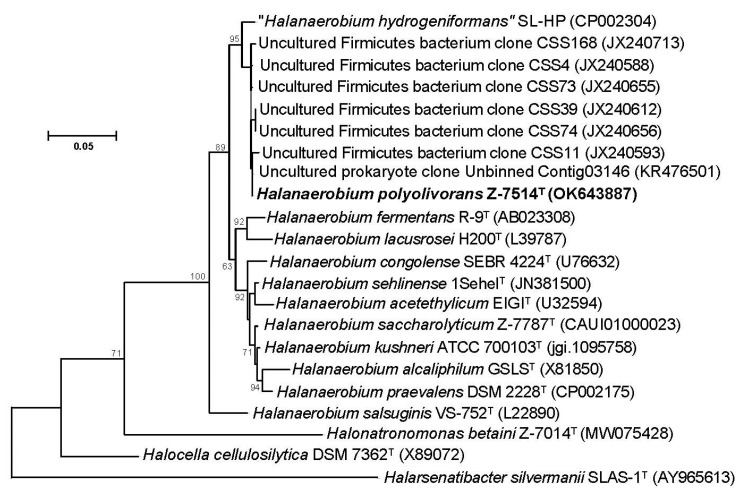
Maximum-likelihood phylogenetic tree based on 16S rRNA gene sequences (1435 nucleotide sites) reconstructed with evolutionary model GTR+I+G4+F, showing the position of strain Z-7514^T^ with closely related members of the genus *Halanaerobium*. Bootstrap values (>50%) are listed as percentages at the branching points. GenBank accession numbers for 16S rRNA genes are indicated in brackets. Bar, 0.05 substitutions per nucleotide position.

**Figure 3 microorganisms-11-02325-f003:**
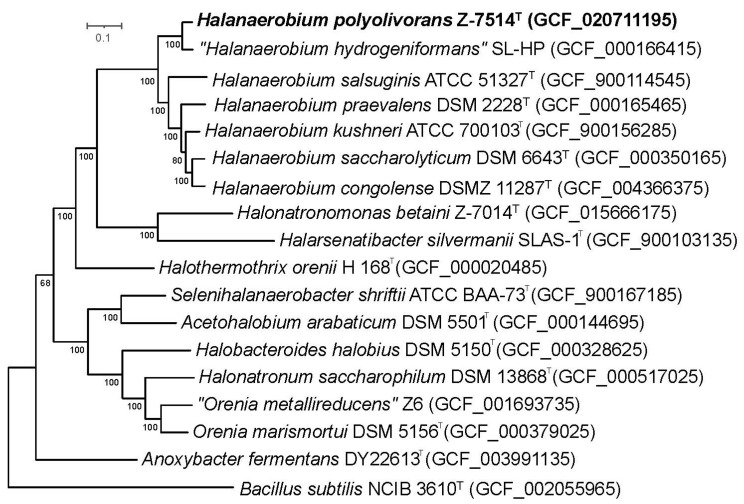
Maximum-likelihood phylogenomic tree derived from concatenated 120 single copy marker proteins showing the position of strain Z-7514^T^ in relation to taxonomically characterized members of the order *Halanaerobiales*. Phylogenomic analysis was performed with an LG+F+I+G4 model based on 34747 amino acid positions. The tree was rooted using *Bacillus subtilis* NCIB 3610^T^ as the outgroup. Accession numbers for the genomes are indicated in brackets. Bar, 0.1 amino acid substitutions per site.

**Figure 4 microorganisms-11-02325-f004:**
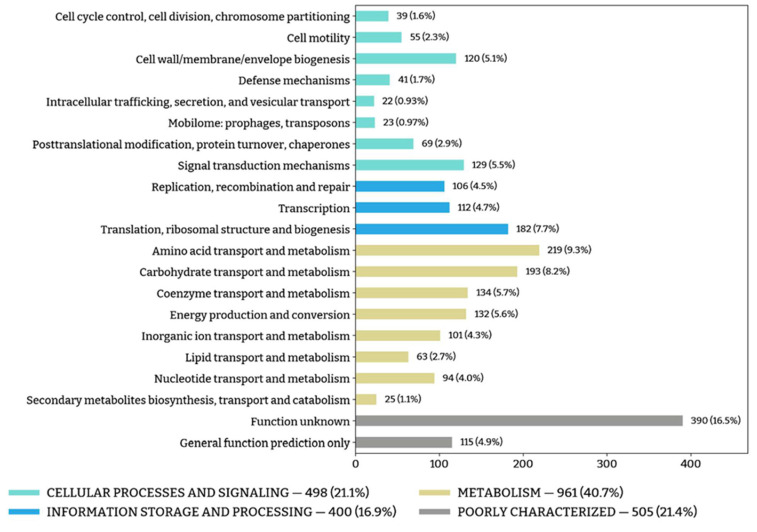
Number of genes associated with the general COG functional categories.

**Figure 5 microorganisms-11-02325-f005:**
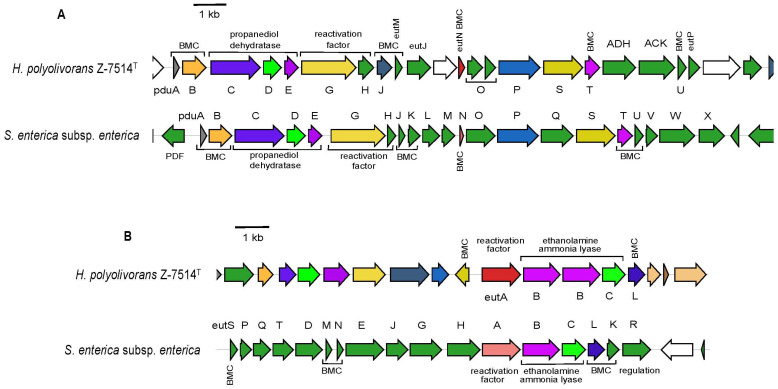
Genetic organization of the *pdu* (**A**) and *eut* (**B**) gene clusters in strain Z-7514^T^ and *Salmonella enterica* subsp. *Enterica* LT2. Colors: red—a pin protein used as a query in the GizmoGene server; white—hypothetical protein; the others—the same color in sequences identifies the protein members of the same family. The functions of some encoded proteins are indicated. The letters between gene sequences refer to *pdu* and *eut* genes, i.e A, B, C, etc., means *pdu*A, *pdu*B, *pdu*C, etc. (panel (**A**)) and *eut*A, *eut*B, *eut*C, etc. (panel (**B**)), respectively. Designations outside sequences are: BMC, bacterial microcompartment; ADH, alcohol dehydrogenase; ACK, acetate kinase; PDF, propanediol diffusion facilitator.

**Figure 6 microorganisms-11-02325-f006:**
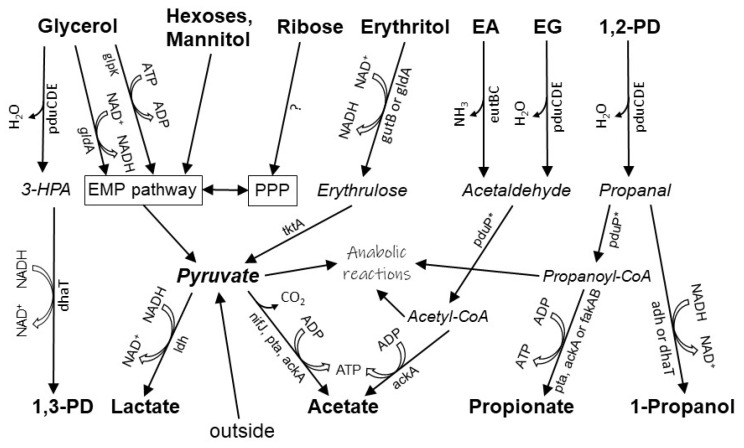
A tentative scheme of substrates’ utilization by strain Z-7514^T^. Substrates and products are marked in bold. Intermediates are marked in italic. Abbreviations for substances: EA—ethanolamine, EG—ethylene glycol, 1,2-PD—propylene glycol, 3-HPA—3-hydroxypropanal, 1,3-PD—1,3-propanediol. Genes and enzymes: *pdu*CDE—1,2-propanediol dehydratase (EC 4.2.1.28), *gld*A—glycerol dehydrogenase (EC 1.1.1.6), *glp*K—glycerol kinase (EC 2.7.1.30), *gut*B—L-iditol (D-sorbitol) 2-dehydrogenase (EC 1.1.1.14), *eut*BC—ethanolamine ammonia lyase (EC 4.3.1.7), *tkt*A—transketolase (EC 2.2.1.1), *pdu*P—propionaldehyde dehydrogenase (EC 1.2.1.87), *dha*T—1,3-propanediol dehydrogenase (EC 1.1.1.202), *ldh*—lactate dehydrogenase (EC 1.1.1.27), *nif*J—pyruvate/ferredoxin oxidoreductase (EC:1.2.7.1), *pta—*phosphate acetyltransferase (EC 2.3.1.8), *ack*A—acetate kinase (EC 2.7.2.1), *fak*AB—nonspecific fatty acid kinase (EC 2.7.2.18). *—NAD^+^ dependent reaction.

**Table 1 microorganisms-11-02325-t001:** Genome statistics of strain Z-7514^T^.

Attribute	Value	% of Total
Genome size (bp)	2,522,622	100.0
DNA coding (bp)	2,266,477	89.9
DNA G+C (bp)	839,862	33.3
DNA scaffolds	53	100.0
Total genes	2457	100.0
Protein coding genes	2364	96.2
RNA genes	63	2.6
Pseudo genes	30	1.2
Genes with function prediction	2215	90.2
Genes assigned to COGs	1974	80.3
Genes with Pfam domains	2116	86.1
Genes with signal peptides	185	7.5
Genes with transmembrane helices	615	25.0
CRISPR repeats	2	-

**Table 2 microorganisms-11-02325-t002:** The genomic indices (%) of strain Z-7514^T^ and some species of the genus *Halanaerobium*.

Species of *Halanaerobium*	dDDH	ANI	AAI	POCP
‘*H. hydrogeniformans*’ SL-HP GCF_000166415.1	29.9	84.8	88.4	86.8
*H. congolense* DSM 11287^T^ GCF_004366375.1	19.2	77.0	71.1	69.0
*H. saccharolyticum* DSM 6643^T^ GCF_000350165.1	18.7	77.2	70.9	72.2
*H. kushneri* ATCC 700103^T^ GCF_900156285.1	18.5	77.0	70.9	64.9
*H. praevalens*_DSM 2228^T^ GCF_000165465.1	18.8	77.0	69.3	63.7
*H. salsuginis* ATCC 51327^T^ GCF_900114545.1	18.8	77.0	68.2	61.9

**Table 3 microorganisms-11-02325-t003:** Carbon balance of some substrates utilized by strain Z-7514^T^.

Substrate	End Products, mol/mol Substrate			Carbon Recovery, %
	Acetate	Lactate	H_2_	
Pyruvate	0.96	0	0.7	96
D-ribose	0.46	0.05	0.6	51
D-glucose	0.53	0.56	0.8	109
D-fructose	0.22	0.93	0.4	115
Erythritol	0.77	0.05	0.9	82
Mannitol	0.38	0.74	0.9	111
Glycerol ^1^	0.46	0.11	7.0	81
Ethanolamine ^2^	1.12	0	3.3	112

^1^ plus 0.24 mol 1,3-propanediol per mol glycerol is formed. ^2^ plus 1.11 mol NH_3_ per mol ethanolamine is formed.

**Table 4 microorganisms-11-02325-t004:** Differential characteristics of strain Z-7514^T^, some close species, and type species of the genus *Halanaerobium*.

	Z-7514^T^	^1^ *H. kushneri* VS-751^T^	^2^ *H. saccharolyticum* Z-7787^T^	^3^ *H. praevalens* GSL^T^
Cell size, µm	0.4–0.8 × 1.0–2.0	0.7 × 2.0–3.3	0.5–0.7 × 1.0–1.5	0.9–11.0 × 2.0–2.6
Na^+^, M, range/opt	0.6–3.9/1.6–2.1	1.5–3.1/2.1	0.5–5.1/1.7	0.3–5.1/2.1
pH, range/opt	6.7–10.1/8.0–8.5	6.0–8.0/6.5–7.5	6.0–8.0/7.5	6.0–9.0/7.0–7.4
T, °C, range/opt	14–51/31–35	20–45/40	15–47/37–40	5–50/37
Utilization of				
L-arabinose	–	+	+	ND
Cellobiose	+	+	+	−
D-galactose	+/−	+	+	−
Glycerol	+	^4^ var	+	−
Lactose	−	+	+	−
Pyruvate	+	+	+	−
Starch	−	−	ND	−
Sucrose	+	+	+	−
D-xylose	+/−	−	+	−
Trypticase	−	+	ND	+
Amino acids	−	−	−	+
N-acetylglucosamine	+	ND	−	+
Fermentation products from sugars	Acetate, lactate, H_2_, CO_2_	Acetate, formate, ethanol, H_2_, CO_2_	Acetate, H_2_, CO_2_	Acetate, formate, lactate, ethanol, H_2_, CO_2_
G+C, mol % from genome	33.3	34.2	34.8	30.3
Habitat	Alkaline sediments of a collector at Tanatar III soda lake, Russia	Hypersaline petroleum reservoir fluid, OK, USA	Hypersaline lagoons of Sivash Lake, Russia	Deep bottom sediment of Great Salt Lake, Utah, USA.

+ means positive, − means negative, ND means not determined. All strains were positive for D-mannose, D-fructose, D-glucose, and maltose utilization and negative for L-sorbose utilization. ^1^ Data are from [18,47]. ^2^ *H. saccharolyticum* subsp. *saccharolyticum* Z-7787^T^ [45]. ^3^ Data are from [16,18]. ^4^ The trait is variable among the strains [47].

## Data Availability

The GenBank/EMBL/DDBJ accession numbers for the 16S rRNA gene sequences, whole genome, and genome assembly of strain Z-7514^T^ are OK643887, JAJFAT000000000, and GCA_020711195, respectively. The type strain of *Halanaerobium polyolivorans*, Z-7514^T^, was deposited in the Korean Collection for Type Cultures (accession number: KCTC 25405) and the All-Russian Collection of Microorganisms (accession number: VKM B-3577).

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
