# Peer review of "Halanaerobium polyolivorans sp. nov.—A Novel Halophilic Alkalitolerant Bacterium Capable of Polyol Degradation: Physiological Properties and Genomic Insights"

_microorganisms, 2023, doi:10.3390/microorganisms11092325_

Round 1
Reviewer 1 Report
This manuscript described on taxonomic studies of novel halophilic novel species belonging to genus Halanaerobium that can degrade polyol.
Major:
1. Although authors described the isolate as haloalkaliphile, the definition of alkaliphile is the microorganisms exhibit the optimum growth pH higher than pH 10. According to the growth characteristics of the isolate, it must classify as alkalitolerant strain.
2. The authors deposit 16S rRNA gene and whole genome sequences separately deposited. Is this mean 16S rRNA gene sequence was determined by amplification specific of the sequence? If so, please describe the method.
3. Please construct phylogenetic tree based on the numerous gene maker sequence.
4. Please describe the reasons for the employment of evolutionary model GTP+I+G4+F.
5. Please analyze fatty acid composition and compared with phylogenetic neighboring strains prepared under the same experimental conditions.
6. Please consider the estimation of average amino acid identity (AAI).
7. Figure 5: Please describe explanation for understanding of readers the reasons depicted as a Figure of genetic organization for genes of pdu and eut in the legend. Please explain the meaning of alphabet and colour of arrows.
8. L339-342: This description should be in “Discussion”.
Author Response
Thank you Reviewer I for valuable notes and comments. Below we have tried to respond all of them. Our answers are in bold.
- – Although authors described the isolate as haloalkaliphile, the definition of alkaliphile is the microorganisms exhibit the optimum growth pH higher than pH 10. According to the growth characteristics of the isolate, it must classify as alkalitolerant strain.
We have reclassified the strain as an alkalitolerant. In the strict sense, there is no internationally accepted definition of alkaliphily and there are quite a few articles describing strains as alkaliphilic with an optimum pH in the 9-10 range.
- – The authors deposit 16S rRNA gene and whole genome sequences separately deposited. Is this mean 16S rRNA gene sequence was determined by amplification specific of the sequence? If so, please describe the method.
This is how it happened historically. When the strain was isolated first, the 16S rRNA gene sequencing was performed to determine the taxonomical position and this sequence was deposited just in case. A year later, when it was decided to prepare an article describing a new species, whole genome sequencing was performed and an assembly was also deposited. Anyway, we have added the technique that was used for 16S rRNA gene sequencing.
- – Please construct phylogenetic tree based on the numerous gene maker sequence.
Done, Fig.3
- – Please describe the reasons for the employment of evolutionary model GTP+I+G4+F.
The information added to the Material and Methods section, GTR+F+I+G4 model was proposed by ModelFinder as the best-fit for our data.
- – Please analyze fatty acid composition and compared with phylogenetic neighboring strains prepared under the same experimental conditions.
We cannot order microbial cultures from the world's collections due to the current sanctions against Russia.
- – Please consider the estimation of average amino acid identity (AAI).
Yes, we did it and have posted in the Table 2.
- – Figure 5: Please describe explanation for understanding of readers the reasons depicted as a Figure of genetic organization for genes of pdu and eut in the legend. Please explain the meaning of alphabet and colour of arrows.
The letters instead of gene names (e.g. C, D, E vs pduC, pduD, pduE, names of the same type) were used to save the space and to guide the reader's eye otherwise, the figure will be difficult to view, bulky. Yes, it looks like an alphabet, but the clusters contain too many genes to “comfort viewing”. In alignments, similar colors mean similar genes (with close homology). The color style is generated by the GizmoGene program (Glob Pfam parameter) and corresponds to that used in the Pfam database. The legend is corrected.
- – L339-342: This description should be in “Discussion”.
Transferred
Reviewer 2 Report
- The English style of the manuscript needs to be improved by an English native editor. Some examples of grammatical mistakes are highlighted in the attached file.
- Please use more attractive keywords instead of too many chemicals names.
- In the abstract, optimal growth conditions should be mentioned.
- Introduction of this manuscript is not similar to ones explaining bacterial taxonomy and nomenclature of new taxon. In this regard, it is recommended to check IJSEM (International Journal of Systematic and Evolutionary Microbiology) articles.
- The unit abbreviation for liter is L (capital letter). Please revise this issue in the whole manuscript.
- Chemotaxonomic markers (peptidoglycan, teichoic acids, and mycolic acids, fatty acids, polar lipids, respiratory quinones, or polyamines) of this strain are not described while they are mandatory in the phenotypic characterization of a novel taxon.
- Phylogenetic tree of a novel taxon should be constructed with valid and type strains of close taxa.
- Due to the introduction of a novel species, the manuscript should have a section entitled “Description of Halanaerobium polyolivorans sp. nov.”.

Please check the attached file.
Author Response
Thank you Reviewer II for valuable notes and comments. Below we have tried to respond all of them. Our answers are in bold.
- The English style of the manuscript needs to be improved by an English native editor. Some examples of grammatical mistakes are highlighted in the attached file.
Thank you very much for this note. We have tried to improve the text.
- Please use more attractive keywords instead of too many chemicals names.
We have slightly changed the list of keywords. Left only two chemical names and added another keyword
- In the abstract, optimal growth conditions should be mentioned.
Inserted.
- Introduction of this manuscript is not similar to ones explaining bacterial taxonomy and nomenclature of new taxon. In this regard, it is recommended to check IJSEM (International Journal of Systematic and Evolutionary Microbiology) articles.
This was a matter of discussion among us. We are preparing an article to this journal for the first time, so we were not sure that the IJSEM style was needed here. But, OK, we have expanded the Introduction in terms of the taxonomic position and short history of the genus and species.
- The unit abbreviation for liter is L (capital letter). Please revise this issue in the whole manuscript.
Done.
- Chemotaxonomic markers (peptidoglycan, teichoic acids, and mycolic acids, fatty acids, polar lipids, respiratory quinones, or polyamines) of this strain are not described while they are mandatory in the phenotypic characterization of a novel taxon.
We know. Unfortunately, we cannot order microbial cultures from the world's collections due to known events around Ukraine. And besides, we cannot transfer payment for strains abroad.
- Phylogenetic tree of a novel taxon should be constructed with valid and type strains of close taxa.
We did it before as you said but, in our previous papers, on the contrary, the reviewers required to include the sequences of the non-valid strains and even uncultured clones in the 16S tree in order to show that isolated strain is not unique in the ecological sense and has taxonomic relatives in the other parts of world. This concept was framed in an article issued this year (https://www.sciencedirect.com/science/article/pii/S0723202022000789). A quote: “To increase the importance of the study, SAM highly recommends screening for the presence and relative abundance of the new taxon in the publicly available 16S rRNA gene amplicon and metagenomic datasets. Read recruitments against the genome using the publicly available metagenomes can reveal relative abundance, biogeographic distribution and ecological patterns that increase the usefulness of the taxonomic description.” For clarity, we have added a short description of the clones that got into the tree when searching the Genbank.
- Due to the introduction of a novel species, the manuscript should have a section entitled “Description of Halanaerobium polyolivorans sp. nov.”.
The section “Conclusions” is renamed into “Description of Halanaerobium polyolivorans sp. nov.” because in fact it is a species description.
Reviewer 3 Report
This manuscript by Boltyanskaya et al., entitled “Halanaerobium polyolivorans sp. nov. – a Novel Haloalkaliphile Capable of Polyol Degradation: Physiological Properties and Genomic Insights”, describes the identification and characterization of a haloalkaliphilic bacterial strain, Z-7514T, that is capable of anaerobically degrading glycols and ethanolamine. The results of several physiological and analytical analyses are described, as well as sequence comparisons between this and related organisms. They conclude that Z-7514T is a novel species, tentatively named Halanaerobium polyolivorans, and it possesses the potentially unique ability to degrade low molecular weight glycols (e.g., ethylene glycol, propylene glycol, glycerol) and the related amino alcohol ethanolamine, with possible applications in environmental remediation.
Generally, this is a good manuscript on an interesting topic. Their experimental approaches are sound and they provide answers for almost all of the essential questions. However, I did have a few concerns that, if addressed, would make for an even better paper suitable for publication in Microorganisms.
[Line 100 and elsewhere] Use of the word “amended” is not correct. In this case and others throughout the manuscript, care proofreading is required.
[Line 152 and elsewhere] In the Results section, the authors describe a wealth of findings regarding H. polyolivorans (e.g., colony morphology). However, a dearth of actual data is present. Perhaps it is not necessary to provide figures for all data described in the text. However, additional data could be provided in the Supplementary Materials files appended to the paper.
[Phylogenetic characterization] While phylogenetic analysis based on the comparison of 16S rRNA genes is standard in the field, I wonder if an alternative phylogenetic analysis (e.g., multi-locus sequence analysis or K-mer profiles) might provide some additional insights. With regards to additional loci, I am not sure what would be most appropriate for Halanaerobium. However, it would need to be sets of genes that are present in related organisms with sequenced genomes.
[Line 191] Where has the complete genome sequence for Halanaerobium polyolivorans been deposited? This was not indicated in the text. Deposition of sequence data as well as its public accessibility is essential before publication.
[3.4.1. Physicochemical Characteristics for Growth] As mentioned previously, the data that led to the findings described in this section should be provided in the Supplementary Materials.
[Table 3, title] Table titles are usually simple sentences. Most of what is written should be in the footnotes.
[Table 3, organization] I would strongly suggest the use of visible cell borders in this table, especially for multi-line values (e.g., Fermentation Products).
[Line 293] Supplementary Table S1 was not provided to this reviewer. It needs to be provided for a proper review.
[3.5. Physiology and Functional Genes] All of these comparisons are based on sequence homologies, correct? However, there is no experimental evidence as to the function of these genes. The absence of expected genes or their substitution by others with alternative expected functionalities does not necessarily mean that a particular pathway is necessarily absent. Strong statements need to be tempered accordingly.
[Figure 5] Comparisons are made between Halanaerobium polyolivorans and Salmonella enterica with regard to their pdu and eut gene clusters. Understandable, given our knowledge of the latter organism. However, I would not be surprised that more closely related Halanaerobium species may also have pdu and eut gene clusters. What is known about them? (n.b. This is discussed after Line 431, thank you.)
[Conclusions] A summary of findings is acceptable. A listing of growth conditions and suitable substrates, not.
There are some unusual word choices sprinkled throughout. Proofreading is necessary.
Author Response
Thank you Reviewer III for valuable notes and comments. Below we have tried to respond all of them. Our answers are in bold.
[Line 100 and elsewhere] Use of the word “amended” is not correct. In this case and others throughout the manuscript, care proofreading is required.
We have tried to improve the English throughout the text.
[Line 152 and elsewhere] In the Results section, the authors describe a wealth of findings regarding H. polyolivorans (e.g., colony morphology). However, a dearth of actual data is present. Perhaps it is not necessary to provide figures for all data described in the text. However, additional data could be provided in the Supplementary Materials files appended to the paper.
It is not entirely clear what the reviewer means, but we have moved Fig. 3 to the Supplementary Materials (Fig. S1 now). Light and electron micrographs are essential for further validation of new taxon and we would like to leave them as is.
[Phylogenetic characterization] While phylogenetic analysis based on the comparison of 16S rRNA genes is standard in the field, I wonder if an alternative phylogenetic analysis (e.g., multi-locus sequence analysis or K-mer profiles) might provide some additional insights. With regards to additional loci, I am not sure what would be most appropriate for Halanaerobium. However, it would need to be sets of genes that are present in related organisms with sequenced genomes.
The same suggestion was made by another reviewer, so we built a phylogenomic tree (and inserted in the text) and calculated the genome indices for more reliably determine the taxonomic position of the strain.
[Line 191] Where has the complete genome sequence for Halanaerobium polyolivorans been deposited? This was not indicated in the text. Deposition of sequence data as well as its public accessibility is essential before publication.
It is. According to the journal’s style this info is at the end of the main text. Named as “Data Availability Statement”, Line 524 in revised manuscript.
[3.4.1. Physicochemical Characteristics for Growth] As mentioned previously, the data that led to the findings described in this section should be provided in the Supplementary Materials.
Done. See the comment above.
[Table 3, title] Table titles are usually simple sentences. Most of what is written should be in the footnotes.
Corrected.
[Table 3, organization] I would strongly suggest the use of visible cell borders in this table, especially for multi-line values (e.g., Fermentation Products).
We are following a journal’s style that does not allow cell borders in the tables.
[Line 293] Supplementary Table S1 was not provided to this reviewer. It needs to be provided for a proper review.
We are very sorry. This is a technical misunderstanding. Uploaded.
[3.5. Physiology and Functional Genes] All of these comparisons are based on sequence homologies, correct? However, there is no experimental evidence as to the function of these genes. The absence of expected genes or their substitution by others with alternative expected functionalities does not necessarily mean that a particular pathway is necessarily absent. Strong statements need to be tempered accordingly.
Yes, indeed, experimental evidences are needed. We will plan to get them later because it will be a separate, biochemical, first of all, work. We have tried to temper some statements in the manuscript so as not to annoy the reader.
[Figure 5] Comparisons are made between Halanaerobium polyolivorans and Salmonella enterica with regard to their pdu and eut gene clusters. Understandable, given our knowledge of the latter organism. However, I would not be surprised that more closely related Halanaerobium species may also have pdu and eut gene clusters. What is known about them? (n.b. This is discussed after Line 431, thank you.)
We have slightly expanded and corrected this paragraph.
[Conclusions] A summary of findings is acceptable. A listing of growth conditions and suitable substrates, not.
The comment is not clear. The content of the “Conclusions” is actually the “Species description”. Another reviewer has proposed to rename it as “Description of Halanaerobium polyolivorans sp. nov.”. We followed the advice because if the paper will be published we intend to send it to the IJSEM for further validation.
Round 2
Reviewer 2 Report
A reference of polyphasic taxonomy is attached for more information.

Author Response
Dear Reviewer,
We believe that the fatty acid profile is of secondary importance at this time. There are numerous examples of variability in this profile depending on culture conditions. The article sent by the reviewer was published in the pre-genomic era (1996) and was relevant in the absence of modern knowledge. Since all physiological functions of a cell are encoded in the genome, in our opinion, a genome comparison (a phylogenomic analysis) of closely related strains better shows the degree of their relationship. Along with the 16S rRNA tree and in accordance with the wishes of another reviewer, we built a whole-genome tree, the analysis of which confirmed the isolated position of the strain. To substantiate our position, I attach an article dated by 2021 and published in the IJSEM.
https://www.microbiologyresearch.org/content/journal/ijsem/10.1099/ijsem.0.005127#tab2
Reviewer 3 Report
The authors have adequately addressed all of my concerns with the revised manuscript. However, with the move of Figure 3 to Supplemental Figure 1, care must be taken to ensure that all subsequent figures are numbered appropriately, both in their legends and in the text. Pending these checks, this manuscript is now suitable for publication.
Generally acceptable.